# WiSparse: Boosting LLM Inference Efficiency with Weight-aware Mixed Activation Sparsity

## Abstract

Large Language Models (LLMs) deliver strong capabilities but incur high inference costs due to dense computation and memory access. Training-free activation sparsity is a promising approach for efficient LLM inference, leveraging its data adaptation and low computational overhead. However, existing methods typically only rely on activation information and a uniform sparsity ratio, overlooking the critical interplay with weights and inter-block sensitivity variation, which leads to suboptimal performance. In this paper, we examine these limitations and identify two key phenomena in modern LLMs: 1) less significant activations may align with highly important weights, and 2) sparsity sensitivity varies non-monotonically across model blocks. To address these issues, we propose a novel **W**eight-aware **Mi**xed-Granularity Training-free Activation **Sparse**ity (**WiSparse**) method that leverages both activation and weight information and enables adaptive sparsity allocation across different granularities. Specifically, we introduce a weight-aware activation sparsification mechanism that integrates activation magnitudes with precomputed weight norms to more accurately identify salient channels. This is combined with a mixed-granularity sparsity allocation scheme featuring a coarse-to-fine strategy: a global sparsity budget is first distributed across blocks via evolutionary search to protect sensitive regions, and subsequently refined at finer granularities within each block to minimize reconstruction error. We improve existing sparse kernels and demonstrate the effectiveness of the proposed method via extensive experiments conducted on three representative models. Notably, at 50% sparsity, WiSparse preserves 97% of Llama3.1's dense model performance, surpassing the strongest baseline by 2.23 percentage points while achieving a 21.4% acceleration in end-to-end inference speed. Our research contributes to advancing the performance limits of training-free approaches for efficient LLM inference, effectively pushing the boundaries of achievable speedup without training.

## 1 Introduction

Large Language Models (LLMs) have demonstrated remarkable capabilities across a wide range of natural language processing tasks, making them fundamental components of modern AI applications (Zhao et al., 2025). However, the extensive parameter of LLMs results in significant computational and memory I/O demands, which severely constrains their inference efficiency. To mitigate this issue, network sparsification has emerged as a promising technique, primarily categorized into weight sparsity and activation sparsity (Wan et al., 2023). Weight sparsity operates in a data-independent manner, where the importance of weights is determined based solely on their intrinsic characteristics, often leading to suboptimal performance in specific scenarios. Consequently, recent research has increasingly focused on data-dependent activation sparsity (Liu et al., 2023; Zhang et al., 2025; Lee et al., 2024; Chen et al., 2025; Liu et al., 2025), which has shown superior performance and better scalability across diverse tasks. Specifically, this work tackles channel sparsity, where entire neuron computations are dynamically pruned based on runtime inputs.

Training-based activation sparsity methods (Song et al., 2024; Wang et al., 2024) improve sparsity through architectural modifications and fine-tuning, yet such methods require substantial computational resources and are highly sensitive to training configurations. These limitations have motivated the development of training-free methods (Lee et al., 2024; Liu et al., 2025; Zhang et al., 2025; Ramachandran et al., 2025), which employ lightweight runtime criteria to identify and bypass less

important computations, preserving the original weights. However, existing training-free sparsity methods predominantly depend on activation-only importance indicators and apply a uniform sparsity ratio across the blocks in the model, thereby neglecting the critical role of weight-activation interactions and the varying sensitivity across different blocks, failing to realize the full potential of activation sparsity.

In this work, we aim to enhance the performance of training-free activation sparsity methods. To this end, we begin by investigating the inherent characteristics of LLMs and identify two central challenges in activation sparsity: (1) *less important activations correspond to critical weights*, and (2) *sparsity sensitivity varies significantly across different blocks*, shown in Sec. 4.1.

To address these challenges, we propose a novel **W**eight-aware **M**ixed-Granularity Training-free Activation **Sparsity** (**WiSparse**) method (see Fig. 1). This approach leverages the joint importance of both weights and activations to guide sparsification, while assigning sparsity ratios according to sensitivity at both the block and layer levels. Specifically, we first propose a weight-aware activation sparsification mechanism to effectively capture and leverage the interactions between weights and activations for computation reduction. To balance the contributions of activation and weight in importance estimation, we introduce a layer-wise exponent parameter, whose optimal value is automatically determined through a lightweight search process tailored to each layer's characteristics. Furthermore, we introduce a mixed-granularity sparsity allocation scheme that progressively assigns sparsity ratios from coarse (block-level) to fine (layer-level) granularities. This is implemented through a two-stage search algorithm: a coarse-grained search first determines the global sparsity distribution across blocks, which is then refined by a fine-grained search that iteratively optimizes layer-level sparsity ratios by minimizing output reconstruction error. Extensive experiments demonstrate that our method outperforms existing baselines, achieving state-of-the-art performance.

In summary, our main contributions are as follows.

- In this work, we identify two critical phenomena in modern LLMs that limit training-free activation sparsity: (1) less significant activations may align with highly important weights, and (2) sparsity sensitivity exhibits non-monotonic variation across different blocks.

- We propose WiSparse, a fully training-free framework that introduces a weight-aware importance score combining both activation and weight information to guide sparsification more accurately. WiSparse further employs a coarse-to-fine search algorithm to optimize sparsity ratios across mixed granularities, adapting to varying sensitivity patterns throughout the model.

- Extensive experiments on diverse LLMs (including Llama3.1, Qwen2.5, and Mistral) demonstrate that WiSparse achieves a superior accuracy-efficiency trade-off. At 50% sparsity, our method not only outperforms representative training-free baselines in accuracy but also accelerates end-to-end inference speed by up to 21.4%.

## 2 RELATED WORK

**LLM Sparsification.** Sparsification has emerged as a mainstream strategy to accelerate LLM inference by reducing the amount of computation required at run time. Early work focused on pruning model parameters to induce weight sparsity (Frantar & Alistarh, 2023; Sun et al., 2023; Yin et al., 2023). Although weight pruning can substantially reduce FLOPs, practical speedups depend on sparse kernel support, memory bandwidth, and sparsity structure. To create more hardware-friendly sparsity, other methods employ *structured* pruning, removing entire components like network modules (Chen et al., 2023) or even full layers (Men et al., 2025), typically followed by a fine-tuning stage to recover performance. A complementary line of work achieves conditional computation through Mixture-of-Experts (MoE) architectures, which instantiate sparsity by activating only a small subset of experts per token (Shazeer et al., 2017; Lepikhin et al., 2020; Fedus et al., 2022). More recently, contextual sparsity has been proposed to skip neuron computations dynamically based on the input context. DejaVu (Liu et al., 2023) exploits the observation that, for ReLU-based networks, many MLP neurons are exactly inactive for a given context and can be predicted cheaply and skipped at run time. However, this approach relies on hard-thresholding properties of ReLU and is not directly applicable to modern LLMs that predominantly adopt smooth activations such as GELU or SwiGLU, where activations are rarely exactly zero.

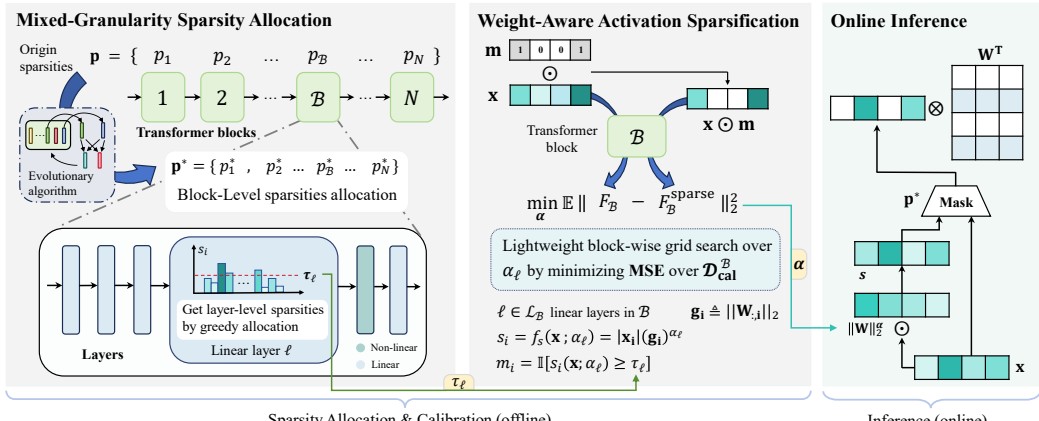

Figure 1: The overall framework of WiSparse. The process starts by calculating importance scores for each layer based on activation values and weight norms. These scores generate sparsity masks to prune less important channels. Block-level sparsity is optimized via an evolutionary search on a small calibration dataset, followed by refinement of layer-level sparsities using a greedy allocation strategy. The final configuration is applied during inference to improve computational efficiency by reducing unnecessary operations.

**Training-based activation sparsity.** To bridge this gap, training-based activation sparsity methods "relufy" or otherwise modify/fine-tune models to encourage sparse activations under smooth non-linearities. Representative approaches include TurboSparse (Song et al., 2024), ProSparse (Song et al., 2025), and Q-Sparse (Wang et al., 2024), modify models by introducing architectural changes, applying sparsity-inducing regularizers, or using specialized fine-tuning. While effective, these methods require nontrivial compute and data to train. Furthermore, their performance is sensitive to the training recipes, and the resulting sparsity-accuracy trade-off may degrade when the model undergoes continual pretraining or encounters domain shifts.

**Training-free activation sparsity.** Training-free activation sparsity methods aim to skip low-importance neuron computations at inference time without any additional fine-tuning, typically by using lightweight, runtime criteria derived from activations. CATS (Lee et al., 2024) performs context-aware thresholding over intermediate activations, enabling dynamic sparsification without modifying model weights or training recipes. It leverages input-dependent statistics to set thresholds so that low-impact activations are skipped while preserving output quality. TEAL (Liu et al., 2025) proposes a training-free framework that applies magnitude-based sparsification to hidden states across layers, together with practical system support to realize real speedups during LLM inference. R-Sparse (Zhang et al., 2025) decomposes computation into a sparse and a low-rank path based on runtime activation magnitudes. High-magnitude activations are processed using the original weights, while the rest are routed through a pre-computed low-rank approximation of the weight matrix. Overall, these training-free approaches base their decisions primarily on activation-side signals and do not explicitly incorporate the underlying weight values into the importance estimation or sparsity scheduling, which may limit their ability to capture weight–activation interactions. WINA (Chen et al., 2025) made an initial step toward incorporating weights by combining activation magnitudes with weight column L2-norms. Although an improvement, this method has two key limitations: first, its use of a simple, static norm is an inadequate proxy for true weight importance, and second, it does not address how to handle mixed sparsity ratios within a model.

## 3 PRELIMINARY

Consider a single linear projection layer $\ell$ in a Transformer block. Let $\mathbf{x} \in \mathbb{R}^{n_\ell}$ and $\mathbf{W} \in \mathbb{R}^{m_\ell \times n_\ell}$ denote the input activation and weight matrix, respectively. The output is

$$\mathbf{y} = \mathbf{x}\mathbf{W}^\top. \tag{1}$$

Activation sparsity seeks, for each input activation, a binary mask vector $\mathbf{m} \in \{0, 1\}^{n_\ell}$ that zeroes out unimportant input channels before the projection:

$$\mathbf{y} = (\mathbf{x} \odot \mathbf{m})\mathbf{W}^\top. \tag{2}$$

Let $S = \{i \in [1 : n_\ell] : \mathbf{m}_i = 1\}$ denote the index of selected channels, and let $k_\ell = |S|$ be the number of such channels. Then only the subvector $\mathbf{x}_S$ and the corresponding columns $\mathbf{W}_{:,S}$ are used:

$$\mathbf{y} = \mathbf{x}_S (\mathbf{W}_{:,S})^\top. \tag{3}$$

This optimization reduces the computational complexity from $O(m_\ell n_\ell)$ to $O(m_\ell k_\ell)$, and proportionally decreases the memory access. The central problem is to construct input-dependent masks $\mathbf{m}$ that achieve a high degree of sparsity with minimal accuracy loss.

## 4 METHOD: WISPARSE

### 4.1 MOTIVATING OBSERVATIONS

We begin by presenting two key observations that reveal the inherent limitations of existing training-free activation sparsity methods and motivate the design of our WiSparse framework.

**Observation 1**: *Less important activations correspond to critical weights.* As shown in Fig. 2, the variance across input channels of the weight matrices is significantly higher than that across output channels, particularly in layers such as o_proj and up_proj. This high variance means that some weight columns have much larger norms than others, rendering pruning decisions based on activation magnitude alone unreliable. For instance, the activation for input channel 2244 is small, but the column of weights associated with this channel possesses one of the highest norms. Consequently, an accurate saliency score must jointly consider both activation and weight information, as activation-only criteria can be misleading.

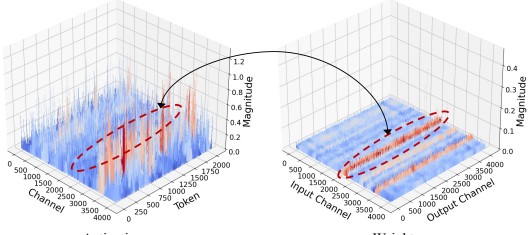

Figure 2: Distribution of activation and weight magnitudes for the self_attn.o_proj layer in block 17 of Llama-3.1-8B. The plot shows that channels with low activation magnitudes can have high-magnitude weights (e.g., channel 2244), demonstrating the limitations of using activation-only metrics to assess channel importance.

**Observation 2**: *Sparsity sensitivity varies significantly across different blocks.* As shown in Fig. 3, sparsifying certain layers in shallow blocks causes markedly larger degradation, whereas deeper blocks are less sensitive and can even yield slight improvements. This non-uniform response demonstrates that blocks have heterogeneous sensitivities. The trend is not strictly monotonic, as some middle blocks also exhibit high fragility, suggesting that a block's criticality depends on its specific functional role and statistical properties rather than just its depth. This observation underscores the sub-optimality of a uniform sparsity policy and highlights the need for an allocation strategy that adapts to the unique sensitivity of each block.

These observations motivate the two core components of our WiSparse framework. To address the inaccurate saliency estimation from Observation 1, we first propose a weight-aware sparsification mechanism. To handle the heterogeneous block sensitivities from Observation 2, we then develop a mixed-granularity allocation strategy. We detail these components in the following subsections.

### 4.2 WEIGHT-AWARE ACTIVATION SPARSIFICATION

To address the issue of misestimated saliency, we explicitly account for the fact that pruning an input channel's impact depends jointly on its activation magnitude and the strength of the corresponding weight column. A simple selection rule is therefore to retain channels with large products $|\mathbf{x}_i| \|\mathbf{W}_{:,i}\|_2$. To better capture weight–activation interactions while keeping a simple, training-free score, we introduce a layer-specific exponent that rescales the weight term. Concretely, given an

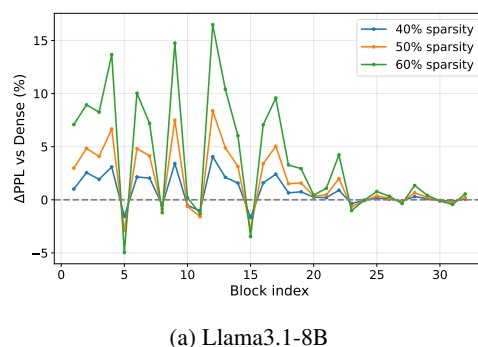 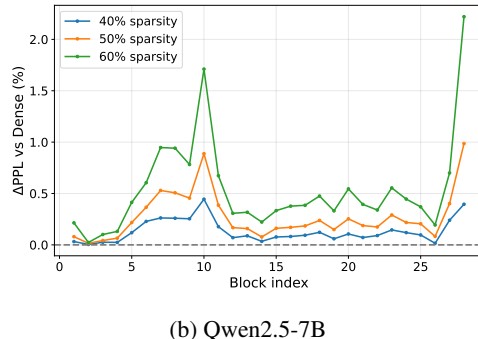

(a) Llama3.1-8B        (b) Qwen2.5-7B

Figure 3: Block-wise sensitivity to sparsification. The plot reports the relative change in validation perplexity ($\Delta$PPL vs the dense model, in %) when sparsifying one block at a time while keeping all other blocks dense. Curves correspond to 40%, 50%, and 60% sparsity.

input $\mathbf{x}$ to layer $\ell$, we define the weight–aware importance score $s_i$ for channel $i$ as

$$s_i = f_s(\mathbf{x}; \alpha_\ell) = |\mathbf{x}_i| (\mathbf{g}_i)^{\alpha_\ell}, \tag{4}$$

where $\mathbf{g}_i \triangleq \|\mathbf{W}_{:,i}\|_2$ is the precomputed L2-norm of the corresponding weight column. Here, $\alpha_\ell$ is a nonnegative, layer-specific exponent tuned on a small calibration set which compresses or accentuates weight-side variability.

At inference time, we apply dynamic sparsity by generating a binary mask $\mathbf{m}$ for each input $\mathbf{x}$. A channel is kept if its importance score $s_i$ exceeds a predetermined, layer-specific threshold $\tau_\ell$:

$$m_i = \mathbb{I}[s_i \geq \tau_\ell], \tag{5}$$

This mask is then applied to the activations before the linear projection. Since $\mathbf{g}_i$ and $\alpha_\ell$ are constant at inference, the per-token overhead is limited to element-wise multiplication, which is negligible compared to the computational savings from the pruned matrix-vector products.

**Tuning Weight Exponents via Block-wise Grid Search** The effectiveness of our method hinges on selecting an optimal weight exponent $\alpha_\ell$ for each layer. To this end, we perform a block-wise grid search as detailed in Alg. 2 in Appendix A.2. For each Transformer block $\mathcal{B}$, which contains a set of linear layers $\mathcal{L}_\mathcal{B}$, we search for the optimal exponents $\{\alpha_\ell\}_{\ell \in \mathcal{L}_\mathcal{B}}$ by minimizing output distortion on a small, dedicated calibration dataset $\mathcal{D}_{\text{cal}}^\mathcal{B}$. This dataset consists of input activations to the block $\mathcal{B}$ collected from a few representative examples. The core of the search involves evaluating candidate exponents $\boldsymbol{\alpha} = \{\alpha_\ell\}_{\ell \in \mathcal{L}_\mathcal{B}}$. For each candidate $\boldsymbol{\alpha}$, we dynamically determine a corresponding set of sparsity thresholds $\boldsymbol{\tau} = \{\tau_\ell\}_{\ell \in \mathcal{L}_\mathcal{B}}$. Each threshold $\tau_\ell$ is calculated to ensure its layer achieves a target keep ratio $r_\ell$ over the block's calibration data.

Let $F_\mathcal{B}$ be the original (dense) forward pass of the block and $F_\mathcal{B}^{\text{sparse}}(\cdot; \boldsymbol{\alpha}, \boldsymbol{\tau})$ be its sparse counterpart. Our objective is to find the exponents that minimize the Mean Squared Error (MSE) between the dense and sparse block outputs, evaluated on the block-specific calibration set:

$$\min_{\boldsymbol{\alpha}} \mathbb{E}_{\mathbf{x}_\mathcal{B} \in \mathcal{D}_{\text{cal}}^\mathcal{B}} \left\| F_\mathcal{B}(\mathbf{x}_\mathcal{B}) - F_\mathcal{B}^{\text{sparse}}(\mathbf{x}_\mathcal{B}; \boldsymbol{\alpha}, \boldsymbol{\tau}(\boldsymbol{\alpha})) \right\|_2^2. \tag{6}$$

During this search, for each $\alpha_\ell$, the threshold $\tau_\ell$ is set as the $(1 - r_\ell)$-quantile of the score distribution and is calculated as:

$$\tau_\ell(\alpha_\ell) = \text{Quantile}_{1-r_\ell}\left( \{s_i(\mathbf{x}_\ell; \alpha_\ell)\} \right). \tag{7}$$

After the grid search identifies the optimal exponents $\{\alpha_\ell^*\}$, we use this same mechanism one final time to compute the fixed, per-layer thresholds for inference. This procedure yields a single, token-agnostic threshold $\tau_\ell$ for each layer. However, since the pruning scores $s_i(\mathbf{x}_\ell; \alpha_\ell)$ depend on the layer's input activations $\mathbf{x}_\ell$, the resulting sparsity pattern is adaptive for each token.

## 4.3 Mixed-Granularity Sparsity Allocation

Our finding that sparsity sensitivity varies significantly across blocks (Sec. 4.1) implies that a uniform allocation strategy is inherently suboptimal. In theory, an optimal design would assign a tailored sparsity ratio to every individual layer. However, exhaustively searching for such a configuration is computationally prohibitive and susceptible to overfitting. To overcome this, we introduce a two-stage, coarse-to-fine allocation strategy that efficiently approximates the optimal per-layer sparsity distribution while maintaining search stability and generalization.

**Coarse Search: Block-Level Allocation** In the first stage, we determine how to distribute a global target sparsity, $p_{\text{target}}$, among the $N$ Transformer blocks of the model. We search for a set of block-level pruning ratios, $\mathbf{p} = \{p_1, \ldots, p_N\}$, where $p_B$ represents the sparsity assigned to block $B$. The allocation is subject to the constraint that the average sparsity across all blocks equals the global target $p_{\text{target}}$.

To find the optimal block-level allocation, we employ an evolutionary algorithm (see Alg. 3 in Appendix A.2). The search process begins by initializing a uniform allocation where every block is assigned the target sparsity $p_{\text{target}}$. In each generation, we create a population of new candidate allocations (offspring) derived from the current best solution. The generation process consists of three key steps:

1. **Localized Mutation:** To create a new candidate, we start with a copy of the parent allocation. We then randomly select a small subset of blocks (e.g., 10% of the total) and increase their sparsity levels by a small, fixed step size $\epsilon$. This localized approach, inspired by (Sieberling et al.), ensures stable exploration of the search space.

2. **Constraint Enforcement:** This mutation may cause the candidate's weighted average sparsity to exceed the global target $p_{\text{target}}$. To maintain the constraint, we then iteratively select random blocks and decrease their sparsity by $\epsilon$ until the average sparsity returns to the target level.

3. **Selection:** After generating a list of offspring, each candidate allocation $\mathbf{p}'$ is evaluated by measuring its performance degradation on a small calibration dataset, $\mathcal{D}_{\text{cal}}$. The candidate that minimizes our objective function $\mathcal{L}(\mathbf{p}')$ is chosen as the parent for the next generation.

Specifically, we use the average token-level KL divergence between the output distributions of the sparse and dense models as our objective function $\mathcal{L}(\mathbf{p})$:

$$\mathcal{L}(\mathbf{p}) = \frac{1}{|\mathcal{D}_{\text{cal}}|} \sum_{\mathbf{x} \in \mathcal{D}_{\text{cal}}} \frac{1}{T(\mathbf{x})} \sum_{t=1}^{T(\mathbf{x})} \text{KL}\left(\text{softmax}(f_\theta(\mathbf{x}_{\leq t})) \,\|\, \text{softmax}(f_{\theta,\mathbf{p}}(\mathbf{x}_{\leq t}))\right) \tag{8}$$

Here, $T(\mathbf{x})$ is the length of the input $\mathbf{x}$, $f_\theta$ and $f_{\theta,\mathbf{p}}$ are the logits produced by the dense and sparse models, respectively, and the divergence is averaged over all the tokens in the calibration set. The evolutionary algorithm seeks to find the set of block-level sparsities $\{\mathbf{p}_B^*\}$ that minimizes this objective. For this stage, we temporarily assume that all layers within a given block $B$ share the same sparsity $p_B$.

**Fine Search: Intra-Block Greedy Allocation** With the optimal block-level sparsities $\{p_B^*\}$ fixed, the second stage refines the sparsity allocation within each block. For each block $B$, we distribute its assigned sparsity budget $p_B^*$ among its individual linear layers (e.g., in the attention and MLP modules).

This is done using a greedy procedure following (Liu et al., 2025). Starting with a fully dense block, we iteratively add a small, fixed increment of sparsity. At each step, we add the sparsity to the layer that causes the smallest increase in the block's output reconstruction error. This process is repeated until the total sparsity of the block reaches its target budget $p_B^*$. This strategy ensures that more sparsity is allocated to less sensitive layers. For the pseudocode, please refer to Alg. 4 in Appendix A.2.

By combining these two stages, our method efficiently determines a well-calibrated, per-layer sparsity distribution that adheres to a global budget while preserving model accuracy.

---

**Algorithm 1** WiSparse Full Pipeline

---

1: **Input:** Model $\mathcal{M}$, calibration data $\mathcal{D}_{\text{cal}}$, target sparsity $p_{\text{target}}$
2: **Output:** Sparse model $\mathcal{M}_{\text{sparse}}$
3: $\mathbf{p}_{\text{block}} \leftarrow$ BlockLevelAllocation$(\mathcal{M}, \mathcal{D}_{\text{cal}}, p_{\text{target}})$    $\triangleright$ Evolutionary search for block sparsities
4: $\mathbf{p}_{\text{layer}} \leftarrow$ IntraBlockAllocation$(\mathcal{M}, \mathbf{p}_{\text{block}}, \mathcal{D}_{\text{cal}})$    $\triangleright$ Greedy search for layer sparsities
5: $\boldsymbol{\alpha} \leftarrow$ SearchAlphaSequences$(\mathcal{M}, \mathcal{D}_{\text{cal}})$    $\triangleright$ Block-wise grid search for each $\alpha_\ell$
6: $\mathcal{M}_{\text{sparse}} \leftarrow \mathcal{M}$
7: **for** each layer $\ell$ in $\mathcal{M}_{\text{sparse}}$ **do**
8:    $\ell$.exponent $\leftarrow \boldsymbol{\alpha}[\ell]$             $\triangleright$ Set layer exponent $\alpha_\ell$
9:    $\ell$.set_threshold$(\mathbf{p}_{\text{layer}}[\ell], \mathcal{D}_{\text{cal}})$     $\triangleright$ Set threshold $\tau_\ell$ using calibration stats
10: **end for**
11: **return** $\mathcal{M}_{\text{sparse}}$

---

Table 1: Accuracy comparison of WiSparse with baseline methods on six tasks. The best performance in each group is shown in **bold**, while the second-best is underlined.

| Model | Sparsity | Method | SIQA | GSM8K | WiC | HumanEval | MMLU | CSQA | Average |
|---|---|---|---|---|---|---|---|---|---|
| | 0 | Baseline | 43.19 | 82.87 | 52.35 | 67.07 | 69.07 | 78.87 | 65.57 |
| | 30 | R-Sparse | 41.40 | 82.56 | 51.41 | **68.90** | **68.45** | 78.21 | 65.16 |
| | | TEAL | 41.71 | **83.85** | 51.25 | 64.63 | 68.01 | **78.54** | 64.67 |
| | | Ours | **43.76** | 82.26 | **52.63** | 67.07 | 68.22 | 78.46 | **65.40** |
| Llama-3.1-8B | 40 | R-Sparse | **42.68** | 80.82 | 50.66 | 61.59 | 66.86 | 77.56 | 63.36 |
| | | TEAL | 41.71 | 82.03 | 50.47 | 63.41 | 67.10 | **78.13** | 63.81 |
| | | Ours | 42.58 | **82.41** | **50.78** | **66.46** | **67.88** | 78.05 | **64.70** |
| | 50 | R-Sparse | 43.04 | 74.98 | **52.35** | 59.76 | 63.43 | 74.46 | 61.34 |
| | | TEAL | 42.73 | 75.51 | 37.30 | 62.20 | 64.71 | 75.27 | 59.62 |
| | | Ours | **43.14** | **78.77** | 50.63 | **65.85** | **65.23** | **77.81** | **63.57** |
| | 0 | Baseline | 66.48 | 55.72 | 58.31 | 37.80 | 61.92 | 73.46 | 58.95 |
| | 30 | R-Sparse | 66.48 | 54.66 | **58.46** | **40.24** | 61.32 | **73.79** | **59.17** |
| | | TEAL | 66.53 | 54.81 | 58.15 | 37.20 | 61.20 | 73.05 | 58.53 |
| | | Ours | **66.79** | **54.89** | 58.31 | 37.80 | **61.60** | 73.46 | 58.76 |
| Mistral-7B | 40 | R-Sparse | **66.48** | 52.62 | 57.05 | 39.02 | 60.25 | 72.65 | 58.00 |
| | | TEAL | 66.38 | 51.71 | 57.52 | 37.80 | 60.14 | 72.89 | 57.71 |
| | | Ours | 66.22 | **53.45** | **57.68** | **39.63** | **60.85** | **73.14** | **58.54** |
| | 50 | R-Sparse | **66.33** | 47.23 | 57.21 | **39.02** | **58.82** | 70.43 | 56.51 |
| | | TEAL | 65.87 | 50.80 | 57.21 | **39.02** | 58.54 | 70.93 | 57.06 |
| | | Ours | 66.22 | **51.71** | **57.52** | 38.41 | 59.52 | **72.81** | **57.70** |
| | 0 | Baseline | 41.45 | 80.67 | 55.02 | 79.87 | 74.26 | 83.87 | 69.19 |
| | 30 | R-Sparse | 41.25 | **81.35** | 54.39 | 76.82 | 73.18 | **82.64** | 68.27 |
| | | TEAL | **42.12** | 79.45 | **55.02** | **81.10** | 71.57 | 81.57 | 68.47 |
| | | Ours | 41.86 | 80.74 | 54.70 | 79.27 | **73.76** | 83.05 | **68.90** |
| Qwen-2.5-7B | 40 | R-Sparse | 40.69 | **80.59** | 53.76 | 79.27 | 72.28 | **82.56** | 68.19 |
| | | TEAL | **41.35** | 78.92 | 53.61 | 81.32 | 71.77 | 81.57 | 68.09 |
| | | Ours | 41.15 | 79.15 | **54.39** | **81.71** | **72.78** | 82.72 | **68.65** |
| | 50 | R-Sparse | 39.82 | 77.86 | 52.19 | 77.44 | 70.68 | 81.57 | 66.59 |
| | | TEAL | 40.12 | 76.27 | 49.84 | 70.73 | 69.55 | 79.77 | 64.38 |
| | | Ours | **40.48** | **79.00** | **52.66** | **78.05** | **71.87** | **81.74** | **67.30** |

## 5 EXPERIMENTS

### 5.1 EXPERIMENTAL SETUP

**Models and Datasets.** In our experiments, we evaluate model sparsification performance on three large language models: Llama-3.1-8B-Instruct (Grattafiori et al., 2024), Mistral-7B-Instruct, Qwen2.5-7B-Instruct (Team, 2024). Sparsity levels are set to 30%, 40%, and 50%, with the original dense model (0% sparsity) serving as the baseline. The evaluation is conducted using the OpenCompass benchmark framework (Contributors, 2023), covering six diverse datasets to assess a wide range of reasoning, commonsense, math, and coding capabilities: SIQA (Sap et al., 2019), GSM8K (Cobbe et al., 2021), WiC (Pilehvar & Camacho-Collados, 2018), HumanEval (Chen et al., 2021), MMLU

(Hendrycks et al., 2020), and CSQA (Talmor et al., 2018). Performance is reported as accuracy for all tasks, and average accuracy across these benchmarks is used to assess overall model capability under different sparsity levels.

**Baselines.** We compare against two representative training-free sparsification baselines R-Sparse (Zhang et al., 2025) and TEAL (Liu et al., 2025) under identical target sparsity levels and the same sparsification scope (all linear layers in the transformer blocks). For fair comparison, no additional fine-tuning or distillation is performed after sparsification and we sparsify only half of the prefilling tokens and all the decoding tokens for all tasks.

**Calibration and Hyperparameters.** We use pile-val (Gao et al., 2020), CodeAlpaca-20k (Chaudhary, 2023) and MetaMathQA (Yu et al., 2023) as the calibration set so that math and code tasks can also be calibrated. For the weight exponent grid search, we iterate through a grid over $[0.0, 1.5]$ with a step size of 0.05. For evolutionary search, we initialize all the blocks with the same target sparsity level. The search runs for 400 iterations, generating 64 offspring in each iteration. Offspring are generated via mutation only, with no crossover. We mutate with a step size of 0.5%, and only 10% of the blocks can be mutated each time to stabilize the search process.

## 5.2 ACCURACY RESULTS

As detailed in Table 3, WiSparse consistently outperforms the training-free baselines, R-Sparse and TEAL, across most tested models and sparsity levels. The results underscore our method's ability to maintain high model accuracy while significantly reducing computational density.

The benefits of our approach are most evident at high sparsity. On Llama-3.1-8B with 50% sparsity, WiSparse retains over 97% of the dense model's average accuracy. This performance significantly surpasses the strongest baselines, outperforming R-Sparse and TEAL by 2.23 and 3.95 percentage points, respectively. The advantage is particularly stark on math and coding tasks such as GSM8K and HumanEval, where WiSparse exhibits markedly less accuracy degradation than its counterparts.

This trend of robust performance holds across other models. On Qwen-2.5-7B at 50% sparsity, WiSparse maintains 97.3% of the original average accuracy and widens its lead over TEAL to nearly 3 percentage points. On Mistral-7B, our method again delivers the highest accuracy among all sparse methods. We do observe that on the small HumanEval benchmark (164 problems), sparse models, including ours, can occasionally score higher than the dense baseline. This is likely a statistical artifact stemming from minor, beneficial changes to computation paths caused by sparsification. WiSparse is also highly effective at moderate sparsity. At a 30% sparsity level, performance degradation is negligible, with an average accuracy drop of just 0.22 percentage points across all models. These comprehensive results validate that WiSparse's weight-aware importance score and mixed-granularity allocation strategy are crucial for preserving model capabilities, effectively pushing the accuracy-efficiency frontier for training-free sparsity.

## 5.3 EFFICIENCY RESULTS

Beyond accuracy preservation, a key goal of WiSparse is to reduce actual computation. To realize these efficiency gains, we extended the high-performance sparse kernels from TEAL (Liu et al., 2025) to incorporate our weight-aware scoring mechanism. We measure efficiency in both theoretical FLOPs and end-to-end inference speed (tokens/s) on a single H20 GPU, generating 200 tokens from a 5-token prompt. Fig. 4 (left) reports achieved TFLOPs. As sparsity increases, FLOPs drop almost linearly, reflecting the skipped activation channels in linear projections. For example, in Llama-3.1-8B, FLOPs decrease from 1.92 TFLOPs at 0% sparsity to 1.03 TFLOPs at 50% sparsity—a 46% reduction. Mistral-7B and Qwen-2.5-7B show similar proportional savings, confirming that our sparsification directly reduces computation. Fig. 4 (right) shows the corresponding throughput gains. For Llama-3.1-8B, speed rises from 153.5 tokens/s (dense) to 179.9 tokens/s (50% sparsity), a 17.2% improvement. Mistral-7B and Qwen-2.5-7B achieve 21.4% and 21.2% faster inference, respectively.

## 5.4 ABLATION STUDIES AND RESULTS ANALYSIS

**Ablation studies.** We conducted an ablation study to validate each component of WiSparse, as shown in Tab. 2. A naive approach using only activation magnitudes for pruning causes a severe

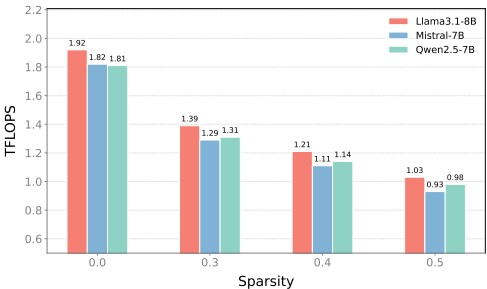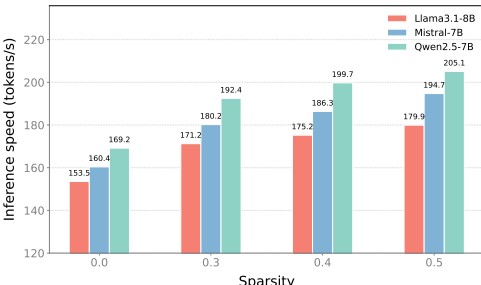

Figure 4: Achieved TLOPS (left) and end-to-end inference speed in tokens/s (right) under different sparsity levels with WiSparse on Llama-3.1-8B, Mistral-7B, and Qwen-2.5-7B.

performance drop to 58.64. Incorporating our weight-aware importance score provides the most significant recovery, boosting accuracy to 61.57 (+2.93). Subsequently, the coarse, block-level search further improves performance to 62.10 by heterogeneously allocating sparsity. Finally, adding the fine-grained, layer-wise search achieves the best result of 63.57. These results confirm that each component is crucial, with the weight-aware importance score providing a strong foundation and the coarse-to-fine search effectively optimizing the final sparsity distribution.

Table 2: **Ablation studies**: the effect of different components proposed in the paper. The experiment is conducted over Llama-3.1-8B on average metrics.

| Method | Sparsity | SIQA | GSM8K | WiC | HumanEval | MMLU | CSQA | Average |
|---|---|---|---|---|---|---|---|---|
| Baseline | 0 | 43.19 | 82.87 | 52.35 | 67.07 | 69.07 | 78.87 | 65.57 |
| Activation only | 0.5 | 42.32 | 74.07 | 42.01 | 56.10 | 63.04 | 74.28 | 58.64 |
| + Weight importance | 0.5 | 43.60 | 76.42 | 50.47 | 60.98 | 63.56 | 74.37 | 61.57 |
| + Coarse search | 0.5 | 42.99 | 76.42 | 51.10 | 62.20 | 64.00 | 75.84 | 62.10 |
| + Fine search | 0.5 | 43.14 | 78.77 | 50.63 | 65.85 | 65.23 | 77.81 | 63.57 |

**Visualization of sparsity allocation.** Fig. 5 visualizes the per-block and per-module sparsity discovered by our coarse-to-fine allocator at a global target of 50% for Llama-3.1-8B and Qwen-2.5-7B. The resulting distributions are clearly heterogeneous across depth and differ between models, reflecting distinct sensitivity profiles. Consistent with Observation 2, the allocator tends to assign lower sparsity to blocks that are empirically fragile and higher sparsity to more robust regions. For example, in Qwen-2.5-7B (Fig. 3b), Block 10 exhibits pronounced sensitivity at 40–60% sparsity; correspondingly, our search assigns it a noticeably lower prune ratio in Fig. 5b. Overall, the learned schedules align with the measured sensitivity landscape (Fig. 3), indicating that the allocator effectively tailors sparsity to model- and block-specific characteristics rather than enforcing a uniform policy.

## 6 CONCLUSION

In this paper, we introduced WiSparse, a novel training-free framework designed to enhance the efficiency of large language model inference through activation sparsity. Our work is motivated by two key observations: less significant activations may align with highly important weights, and the heterogeneous sensitivity of transformer blocks to sparsification. WiSparse addresses these challenges by incorporating a weight-aware importance score that jointly evaluates activation magnitudes and weight norms, and by employing a coarse-to-fine allocation strategy that tailors sparsity ratios at both the block and layer levels. Our methodology requires no model retraining and incurs negligible runtime overhead. Extensive experiments on models like Llama-3.1-8B and Qwen-2.5-7B demonstrate that WiSparse consistently outperforms existing training-free baselines, retaining over 97% of dense model accuracy at 50% sparsity while delivering substantial throughput gains.

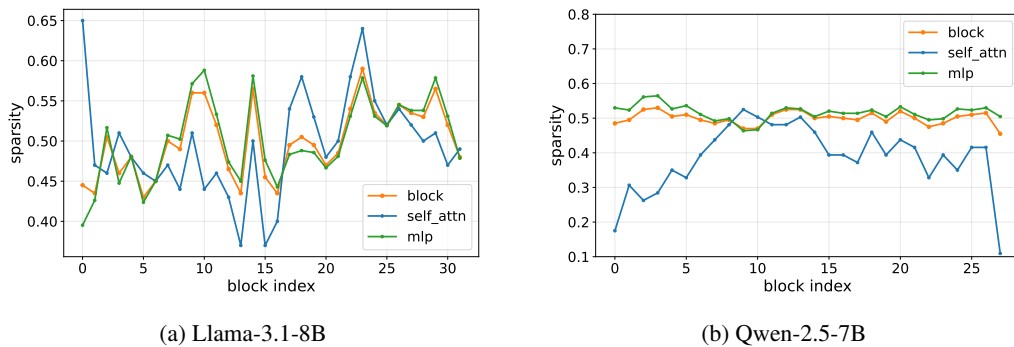

(a) Llama-3.1-8B                 (b) Qwen-2.5-7B

Figure 5: Per-block and per-module (self-attention and MLP) sparsity distributions for (a) Llama-3.1-8B and (b) Qwen-2.5-7B, as determined by our search algorithm targeting 50% overall sparsity.

**Limitations and future work.** The primary limitation stems from the dynamic, token-dependent nature of the masks. While this adaptivity is key to preserving accuracy, it can introduce runtime overhead from mask computation and may complicate efficient implementation for batched inference, where each sequence can yield a different sparsity pattern. Future research could therefore focus on optimizing kernels for dynamic structured sparsity, especially in a batch setting. Additionally, the offline calibration relies on a search procedure; developing more efficient or analytical methods for this step would enhance the framework's practicality and reduce setup costs.

## 7 ETHICS STATEMENT

This research aims to contribute to societal well-being by developing methods that make LLMs more computationally efficient, thereby increasing their accessibility and reducing their environmental impact. The work adheres to high standards of scientific excellence through a transparent methodology, comprehensive experiments, and detailed reporting of results to ensure reproducibility. We have respected the work of others by building upon and citing prior research. All models and datasets used in this study are publicly available, and their use is consistent with their original licenses, with no private or sensitive user data being involved. We have strived for honesty and transparency by clearly outlining our methods, limitations, and the potential impacts of our work, including a disclosure on the use of LLMs as an editorial tool in the preparation of this manuscript.

## 8 REPRODUCIBILITY STATEMENT

We commit to ensuring the reproducibility of our work. The core components of our method, including the weight-aware importance score and the mixed-granularity allocation strategy, are detailed in Sec. 4. Furthermore, Appendix A.2 provides complete pseudocode for the block-wise grid search, evolutionary block-level allocation , and greedy layer-level allocation. Sec. 5.1 details our experimental setup, specifying the models, evaluation benchmarks, calibration data, and all necessary hyperparameters to replicate our results.

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

# A APPENDIX

## A.1 THE USE OF LARGE LANGUAGE MODELS (LLMS)

Throughout the preparation of this manuscript, LLMs were employed as editorial tools. We used them for tasks such as grammar and style editing, and enhancing clarity and concision. The scientific concepts, study design, data analysis, and conclusions were conceived and developed exclusively by the human authors. LLMs were used solely to refine the wording of ideas already established. We provide this disclosure to maintain transparency and uphold academic integrity and responsible research practices.

## A.2 ALGORITHMS

---

**Algorithm 2** Lightweight Block-Wise Grid Search for Alpha

---

**Input:** Block $\mathcal{B}$, linear layers $\ell$ and weight matrix $W$, input vector $X$
**Output:** Optimal scaling factors $\alpha^*$
org_out $\leftarrow$ BlockForward($\mathcal{B}, X,$ None)                              $\triangleright$ Original output
score $\leftarrow$ GetWeightScale($W$)                                        $\triangleright$ Weight importance scores
best_error $\leftarrow +\infty$, best_ratio $\leftarrow -1$, best_scales $\leftarrow$ None
$n_{\text{grid}} \leftarrow 30$, history $\leftarrow [\,]$                          $\triangleright$ Grid search resolution
org_state_dict $\leftarrow \mathcal{B}$.state_dict()
**for** ratio $\leftarrow 0$ **to** $n_{\text{grid}} - 1$ **do**
    $\alpha \leftarrow$ ratio $\times 1.5/n_{\text{grid}}$                              $\triangleright$ Grid point in $[0, 1.5]$
    scales $\leftarrow$ score$^\alpha$.clamp(min $= 1e-4$)                   $\triangleright$ Compute scaling factors
    out $\leftarrow$ BlockForward($\mathcal{B}, X,$ scales)
    loss $\leftarrow \|\text{org\_out} - \text{out}\|_2^2$                              $\triangleright$ Mean squared error
    history.append(loss)
    **if** loss $<$ best_error **then**
        best_error $\leftarrow$ loss, best_ratio $\leftarrow \alpha$, best_scales $\leftarrow$ scales
    **end if**
    $\mathcal{B}$.load_state_dict(org_state_dict)                              $\triangleright$ Restore original state
**end for**
**return** best_scales

---

**Algorithm 3** Block-Level Sparsity Allocation (Evolutionary Algorithm)

---

**Input:** Model $\mathcal{M}$ with $N$ blocks, target sparsity $p_{\text{target}}$, max generation $G_{\max}$, step size $\epsilon$
**Output:** Optimal block sparsities $\mathbf{p}_{\text{block}}^* = \{p_B^* : B \in \mathcal{M}\}$
$\mathbf{p} \leftarrow \{p_{\text{target}} : B \in \mathcal{M}\}$                  $\triangleright$ Initialize uniform block sparsities
**for** generation $\leftarrow 1$ **to** $G_{\max}$ **do**
    offspring_list $\leftarrow [\,]$
    **while** |offspring_list| $< N_{\text{offspring}}$ **do**
        $\mathbf{p}' \leftarrow \mathbf{p}$.copy()
        num_flips $\leftarrow \lfloor N \times 0.1 \rfloor$                      $\triangleright$ Localized mutation
        **for** $i \leftarrow 1$ **to** num_flips **do**
            $B_{\text{incr}} \leftarrow$ RandomChoice($\{1, \ldots, N\}$)
            $p'_{B_{\text{incr}}} \leftarrow p'_{B_{\text{incr}}} + \epsilon$
        **end for**
        **while** WeightedAverage($\mathbf{p}'$) $> p_{\text{target}}$ **do**        $\triangleright$ Maintain global constraint
            $B_{\text{decr}} \leftarrow$ RandomChoice($\{1, \ldots, N\}$)
            $p'_{B_{\text{decr}}} \leftarrow p'_{B_{\text{decr}}} - \epsilon$
        **end while**
        offspring_list.append($\mathbf{p}'$)
    **end while**
    $\mathbf{p} \leftarrow \arg\min_{\mathbf{p}' \in \text{offspring\_list}} \mathcal{L}(\mathbf{p}')$   $\triangleright$ Select best candidate through objective function
**end for**
**return** $\mathbf{p}$

---

**Algorithm 4** Layer-Level Greedy Sparsity Allocation

---

**Input:** Block $\mathcal{B}$, target block sparsity $p_{\mathcal{B}}^*$, step size $\delta$,
      input activations $\mathbf{X}$, target activations $\mathbf{Y}$
**Output:** Optimal layer sparsities $\mathbf{p}_{\text{layer}}^* = \{p_\ell^* : \ell \in \mathcal{B}\}$
$\mathbf{p} \leftarrow \{p_\ell : 0.0 \text{ for } \ell \in \mathcal{B}\}$               ▷ Initialize layer sparsities to zero
**while** EffectiveSparsity($\mathbf{p}$) $< p_{\mathcal{B}}^*$ **do**
    best_error $\leftarrow +\infty$, best_layer $\leftarrow$ None
    **for** $\ell \in \mathcal{B}$ **do**
        **if** $p_\ell \geq 1.0$ **then continue**
        **end if**                    ▷ Skip if fully sparse
        $\mathbf{p}' \leftarrow \mathbf{p}.\text{copy}()$
        $p_\ell' \leftarrow p_\ell' + \delta$             ▷ Increment layer sparsity
        SetBlockSparsities($\mathcal{B}, \mathbf{p}'$)
        $\mathbf{Y}' \leftarrow$ BlockForward($\mathcal{B}, \mathbf{X}$)
        error $\leftarrow \|\mathbf{Y} - \mathbf{Y}'\|_2^2$      ▷ Block output reconstruction error
        **if** error $<$ best_error **then**
            best_error $\leftarrow$ error, best_layer $\leftarrow \ell$
        **end if**
    **end for**
    $p_{\text{best\_layer}} \leftarrow p_{\text{best\_layer}} + \delta$       ▷ Update layer with lowest error
**end while**
**return** $\mathbf{p}$

---

## A.3 ADDITIONAL RESULTS

Table 3: Additional accuracy comparison of WiSparse with baseline methods on six tasks.

| Model | Sparsity (%) | Method | SIQA | GSM8K | WiC | HumanEval | MMLU | CSQA | Average |
|---|---|---|---|---|---|---|---|---|---|
| | 0 | Baseline | 43.19 | 82.87 | 52.35 | 67.07 | 69.07 | 78.87 | 65.57 |
| | 30 | TEAL | 41.71 | 83.85 | 51.25 | 64.63 | 68.01 | 78.54 | 64.66 |
| | | R-Sparse | 41.40 | 82.56 | 51.41 | 68.90 | 68.45 | 78.21 | 65.15 |
| | | WINA | 41.12 | 84.00 | 51.25 | 65.85 | 68.10 | 78.64 | 64.82 |
| | | **Ours** | **43.76** | 82.26 | **52.63** | 67.07 | 68.22 | 78.46 | **65.40** |
| Llama3.1-8B | 50 | TEAL | 42.73 | 75.51 | 37.30 | 62.20 | 64.71 | 75.27 | 59.62 |
| | | R-Sparse | 43.04 | 74.98 | **52.45** | 59.76 | 63.43 | 74.46 | 61.35 |
| | | WINA | 42.78 | 77.44 | 50.47 | 63.41 | 64.82 | 75.79 | 62.45 |
| | | **Ours** | **43.14** | 78.77 | 50.63 | 65.85 | 65.23 | 77.81 | 63.57 |
| | 65 | TEAL | 48.52 | 32.98 | **45.77** | 22.56 | 46.65 | 57.99 | 42.41 |
| | | R-Sparse | 50.56 | 30.17 | 38.40 | 17.68 | 46.62 | 55.94 | 39.90 |
| | | WINA | 51.33 | 34.99 | 38.64 | 25.61 | 48.24 | 58.20 | 42.84 |
| | | **Ours** | 51.64 | 41.17 | 39.97 | 32.93 | 50.37 | 60.44 | 46.08 |
| | 0 | Baseline | 32.45 | 83.02 | 56.58 | 89.02 | 83.36 | 86.57 | 71.83 |
| | 30 | TEAL | 32.18 | 82.71 | 56.27 | 88.41 | 83.12 | 86.31 | 71.50 |
| | | R-Sparse | 32.35 | 82.48 | 56.42 | 88.41 | 83.05 | 86.19 | 71.48 |
| | | WINA | 32.45 | 82.87 | 56.90 | 88.41 | 83.31 | 86.57 | 71.75 |
| | | **Ours** | 32.61 | 82.64 | 56.74 | 89.02 | 83.18 | 86.42 | **71.76** |
| Qwen2.5-32B | 50 | TEAL | 33.89 | 77.23 | 55.81 | 86.59 | 80.47 | 84.73 | 69.78 |
| | | R-Sparse | 34.12 | 76.89 | 56.11 | 85.98 | 79.88 | 84.21 | 69.53 |
| | | WINA | 34.80 | 77.56 | 56.43 | 87.20 | 80.93 | 85.08 | 70.33 |
| | | **Ours** | 34.52 | 78.34 | 56.27 | 87.80 | 81.41 | 85.51 | 70.64 |
| | 65 | TEAL | 36.24 | 48.67 | 52.83 | 68.90 | 72.15 | 76.34 | 59.18 |
| | | R-Sparse | 37.13 | 45.81 | 51.26 | 65.24 | 71.37 | 74.88 | 57.61 |
| | | WINA | 35.77 | 50.23 | 47.96 | 70.73 | 76.41 | 77.19 | 59.72 |
| | | **Ours** | 36.88 | **54.46** | **53.17** | 75.61 | 77.84 | 79.52 | **62.91** |

## A.4 Layer-wise Visualization of $\alpha$ values

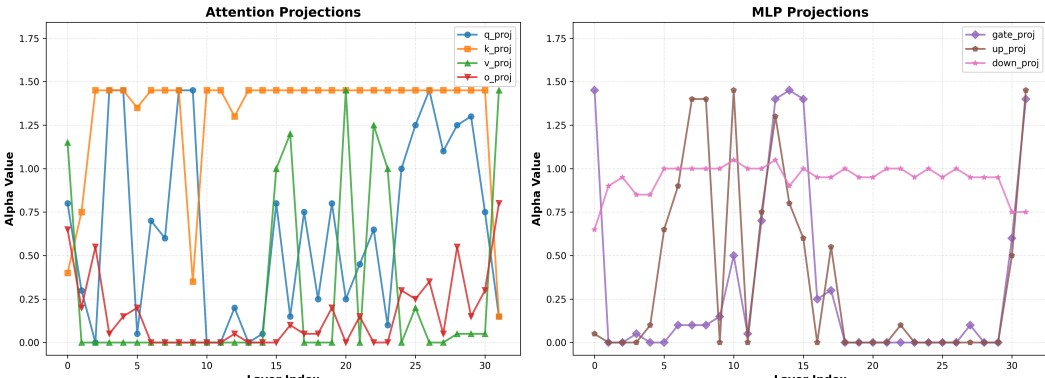

Figure 6: Visualization of the calibrated $\alpha$ across model layers. The left panel displays the $\alpha$ values for attention projection matrices, while the right panel shows them for MLP projection matrices. The x-axis represents the layer index.

