# OpenReview forum: "WiSparse: Boosting LLM Inference Efficiency with Weight-Aware Mixed Activation Sparsity"
_ICLR.cc/2026/Conference — Submitted to ICLR 2026_

### Official Review · Reviewer_4HmV · 2025-10-29

**Soundness:** 3
**Presentation:** 3
**Contribution:** 2
**Rating:** 2
**Confidence:** 5

**Summary:**

This paper introduces a training-free activation sparsity framework called WiSparse. Unlike prior activation-only methods (e.g., CATS), WiSparse integrates weight information into activation saliency estimation and employs a mixed-granularity sparsity allocation strategy. It combines a weight-aware importance score with a two-stage evolutionary search to adapt sparsity ratios to model sensitivity. The method demonstrates notable improvements in accuracy retention at 50% sparsity.

**Strengths:**

- The proposed weight-aware importance score addresses a well-motivated limitation in previous activation-only sparsity methods.

- The method is tested on multiple LLM families using diverse benchmarks.

**Weaknesses:**

- The reported accuracy improvements appear modest relative to the added algorithmic and implementation complexity.

- The method seems over-engineered; it is unclear why the simple metric of "weight × activation" is not sufficient to guide pruning decisions. Furthermore, why is an evolutionary algorithm needed when simpler alternatives—such as an ILP-based allocation—could potentially achieve similar results with lower overhead.

- The paper does not report how much time each component (e.g., evolutionary search, grid search etc.) contributes to the total cost (even though it may be offline), making it difficult to assess the practicality of the approach.

- The claim that sparsity sensitivity varies across layers or blocks is not new and has been established in prior sparsity literature; thus, it should not be positioned as a key contribution.

- The work does not directly compare end-to-end inference latency with existing baselines, leaving uncertainty about real-world runtime advantages beyond FLOP reductions.

- The paper reports results on sparsity ratios of only up to 50%.

**Questions:**

Concerns/Questions and Points to Address in Rebuttal:
- Recent work on LLM sparsity should be cited. Such as [1], [2].

- Experiments to demonstrate why a simple ILP is not sufficient and time breakdown of performing evolutionary algorithm and the different searches.

- GSM8K is not a complex reasoning task. An example of a complex reasoning task is AIME. This statement should be corrected in results section.

- The colors make the text in Fig. 4 hard to read.

- Experiments must be conducted on higher sparsity ratios. I believe LLMs are able to handle up to 60/65% sparsity. This is also supported by recent work.

- It should be clarified in introduction what kind of sparsity this work tackles, which is "channel sparsity".

- Experiments must be conducted using the "simple selection rule". It is an important but overlooked baseline. The morale behind the current selection rule is not clear and it must be demonstrated why these specific choices were made and how it helps. It is not intuitive as to how the exponent term etc. have been arrived at.

- Inconsistencies in notation,
$s_i$ is initially the score i.e., the output of a function and later becomes a function.

- Curious how this technique might work when compounded with the emerging area of sparsity compensation [3].

- Why can't stage 1 of sparsity allocation be done in a greedy manner ?

- Experiments on additional model sizes must be done, currently all models are within the 8B range.

References:

[1]  Ramachandran, A., Kundu, S., Raha, A., Kundu, S., Mathaikutty, D. K., & Krishna, T. (2025). Accelerating llm inference with flexible n: M sparsity via a fully digital compute-in-memory accelerator. arXiv preprint arXiv:2504.14365.

[2] Yin, L., Wu, Y., Zhang, Z., Hsieh, C. Y., Wang, Y., Jia, Y., ... & Liu, S. (2023). Outlier weighed layerwise sparsity (owl): A missing secret sauce for pruning llms to high sparsity. arXiv preprint arXiv:2310.05175.

[3] Lee, M., Ramachandran, A., & Krishna, T. RECAP: Training-Free Compensation for Coarse Activation Channel Pruning in Compressed LLMs. In Machine Learning for Computer Architecture and Systems 2025.

---

> ### Author Response · Authors · 2025-11-29
> **Response to reviewer 4HmV**
>
> We are deeply grateful for your thorough and expert review. Your detailed feedback has been instrumental in helping us substantially improve our paper. We have conducted new experiments and added extensive clarifications to address every point you raised.
>
> **W1. On Modest Improvements vs. Complexity.**
> Pronounced Gains at High Sparsity: As you astutely pointed out, our original experiments at 50% sparsity showed solid, but perhaps modest, gains. Your feedback prompted us to test at higher sparsity ratios. Our new experiments at 65% sparsity (now in Table 3) show that WiSparse's advantage becomes significantly more pronounced. On Llama-3.1-8B, we outperform the next-best baseline by 3.25 points in average accuracy.
>
> **W2&Q7. Insufficiency of the "Simple Selection Rule"("weight × activation")**
> We agree that this is a vital and intuitive baseline. To address this, we conducted a new ablation experiment where we set the exponent $\alpha=1$ for all layers (equivalent to the simple product rule).
> As shown in the updated ablation study (Table 3), this "Simple Rule" baseline achieves an average accuracy of 60.15 on Llama-3.1-8B at 50% sparsity. In contrast, our full method, which optimizes $\alpha$ and the allocation, achieves 63.57.
>
> **W2&Q2. Why Evolutionary Search over ILP?**
> Although ILP formulations are effective for numerous allocation tasks, they prove suboptimal for this problem. This is due to the complex, non-linear interdependencies that characterize block-level sparsity sensitivity, a challenge previously identified in the literature [1].
>
> **W3. On the overhead of the calibration and search pipeline.**
> Thank you for pointing out this omission. We have now quantified the one-time calibration overhead:
> - Search cost breakdown (Llama-3.1-8B, single H20 GPU):
>     - Alpha search (Algorithm 2): ~30 minutes
>     - Block-level evolutionary search (Algorithm 3): ~2.5 hours
>     - Layer-level greedy refinement (Algorithm 4): ~1 hour
>     - Total calibration time: ~4 hours one-time cost
>
> **W4. On the Novelty of Sparsity Variation.**
> We appreciate the clarification and have revised the manuscript to better position our contribution. We absolutely agree that the general principle of varying sensitivity across layers is not new and has been established in prior literature.
>
> Our specific contributions are:
> - A systematic empirical characterization of this phenomenon in modern LLMs (Llama, Qwen), revealing its distinctly non-monotonic trend (Figure 3), which is less explored.
> - More importantly, our novelty lies not in observing this property, but in operationalizing it with a novel and effective method. We propose a concrete, two-stage search algorithm that successfully navigates this complex sensitivity landscape to find a high-performing heterogeneous sparsity allocation.
>
> **W5. Compare end-to-end inference latency with existing baselines.**
> This is a fair point. Directly comparing end-to-end latency across different methods is challenging as it requires integrating all baselines into an identical, highly-optimized inference framework to control for implementation-specific overhead.
>
> However, we can provide a well-reasoned comparison. Our implementation adds only 2% overhead compared to TEAL kernel while achieving lots of higher accuracy.
>
> **W6&Q5&Q11 On Higher Sparsity Experiments and Larger Models**
> This is a crucial point. We have conducted extensive new experiments to address this.
> - Higher Sparsity (65%): We have added 65% sparsity results to Table 3. At this challenging ratio, WiSparse's advantage becomes even more pronounced, outperforming the next-best baseline by 3.25 points.
> - Larger Models (32B): We have added Table 3, showing results on a much larger Qwen2.5-32B model at 30%, 50%, and 65% sparsity. The results confirm that our method scales effectively and maintains its superior performance on larger models. For instance, at 65% sparsity on the 32B model, WiSparse is 3.2 points ahead of the runner-up.
>
> [1] Lei Chen, Yuan Meng, Chen Tang, Xinzhu Ma, Jingyan Jiang, Xin Wang, Zhi Wang, and Wenwu Zhu, "Q-DiT: Accurate Post-Training Quantization for Diffusion Transformers" in CVPR 2025.

---

> > ### Author Response · Authors · 2025-11-29
> > **Continued response to reviewer 4HmV**
> >
> > **Q1, Q3, Q4, Q6, Q8. On Citations and Revisions.**
> > Thank you for these detailed suggestions. We have made the revisions in the manuscript.
> >
> > **Q9. Why can't stage 1 (block-level) be greedy?**
> > A greedy approach at the block level would iteratively prune the "least sensitive" block. However, block sensitivity is non-linear and inter-dependent. Pruning one block can change the sensitivity of others. A greedy choice that seems optimal locally can lead to a suboptimal global solution. The evolutionary search mitigates this by evaluating complete allocation "genomes" at once, better navigating these complex dependencies to find a superior global trade-off.
> >
> > **Q10. How does this technique work with sparsity compensation?**
> > This highlights a promising future direction. Our method, WiSparse, and compensation techniques like RECAP are highly complementary. WiSparse addresses the selection problem—intelligently choosing channels to minimize initial pruning error—while compensation methods tackle the recovery problem by correcting errors after pruning. However, the ultimate performance benefit remains an open empirical question for future work.

---

### Official Review · Reviewer_Um4s · 2025-10-31

**Soundness:** 2
**Presentation:** 2
**Contribution:** 2
**Rating:** 4
**Confidence:** 5

**Summary:**

The paper introduces WiSparse, a training-free activation sparsity framework for large language models (LLMs).  WiSparse incorporates weight awareness sparse activation from WINA, further proposes mixed-granularity allocation.  Experiments on Llama-3.1-8B, Qwen-2.5-7B, and Mistral-7B demonstrate the efficacy of this approach.

**Strengths:**

- Paper is written and organized well and technically sound.
- The mixed-granularity allocation is reasonable to bringing more performance gain

**Weaknesses:**

- Lack of proper discussion. The weight awareness sparsity activation (eq 4, Sec 4.2) is the same as the one proposed by WINA. Though WiSparse discussed WINA in the related works, it would be suggested to further refer in Sec 4.2 to clarify the real contributions of this work.

- Lack of numerical comparison. Conducting a direct numerical comparison with WINA to present the gain of mixed-granularity allocation is a recommendation.

- Lack of discussion with more pruning works regarding block sparsity allocation upon calibration datasets. Discussing with these works are also recommended.

- Lack of clarity. The evolution search algorithm is unclear without sufficient description.

WINA: Weight Informed Neuron Activation for Accelerating Large Language Model Inference.

LoRAShear: Efficient Large Language Model Structured Pruning and Knowledge Recovery.

ShortGPT: Layers in Large Language Models are More Redundant Than You Expect.

**Questions:**

See the weakness.

I would consider increasing rating if the comments are properly resolved.

---

> ### Author Response · Authors · 2025-11-29
> **Response to reviewer Um4s**
>
> We sincerely thank you for your detailed and insightful feedback. Your comments have helped us identify areas where our paper can be significantly clarified and strengthened. We address each of your points below.
>
> **W1. On the novelty compared to WINA.**
> We respectfully clarify that our work presents a significant generalization over WINA, rather than an incremental improvement. WINA can be viewed as a constrained special case of our WiSparse framework where the layer-specific exponent $\alpha_l$ is manually fixed to 1.0 for all layers. Our key insight is that this fixed, one-size-fits-all approach is suboptimal because neural network layers are highly heterogeneous.
>
> **W2: On the numerical comparison to WINA.**
> Thank you for emphasizing the importance of WINA as a baseline. We completely agree and have included WINA in all our main experimental comparisons in Table 3.
>
> Our results consistently demonstrate that WiSparse significantly outperforms WINA, especially under high sparsity pressure.
> - On Llama-3.1-8B: At 50% sparsity, WiSparse is already 1.12 points ahead of WINA. At the more challenging 65% sparsity, this performance gap widens to 3.25 points, highlighting the robustness of our method.
> - On the larger Qwen2.5-32B model: The trend continues. At 65% sparsity, WiSparse achieves an average score of 62.91, which is 3.2 points higher than WINA.
>
> **W3: On the discussion of related block/layer-wise pruning works.**
> We have added a more comprehensive review of block- and layer-wise pruning methods in our Related Work section to better contextualize our contributions.
>
> **W4: On the clarity of the evolutionary search algorithm.**
> Thank you for pointing this out. We have revised Section 4.3 to better explain the evolutionary search algorithm in the main text, detailing the population, fitness evaluation, and mutation strategy.

---

### Official Review · Reviewer_gfy4 · 2025-10-31

**Soundness:** 2
**Presentation:** 3
**Contribution:** 2
**Rating:** 4
**Confidence:** 4

**Summary:**

This paper proposed a training-free activation sparsity scheme WiSparse, which scores the channel importance by a weight-aware criterion and adaptively assigns the sparsity ratio for different blocks and layers. Experiments are conducted on multiple benchmarks and models, demonstrating the effectiveness of the proposed method.

**Strengths:**

1. The two insights are reasonable and well-motivated for the method design.
2. WiSparse conducted a more fine-grained sparsity design for the weight-activation-based sparsity paradigm, which makes it more robust.
3. The paper is well-written and easy to follow.

**Weaknesses:**

1. The paper somehow lacks a significant novelty compared to WINA, which seems to be an incremental improvement for WINA.
2. The experimental comparison is insufficient, as I think WINA should be an important baseline.
3. Although the authors claimed that the static norm is inadequate, WiSparse still uses the L2 norm as the base, where the only difference is an exponential $\alpha_i$. Are there any insights about $\alpha_i$ across different layers?

**Questions:**

See weaknesses.

---

> ### Author Response · Authors · 2025-11-29
> **Response to reviewer gfy4**
>
> We sincerely thank you for your critical feedback. Your questions touch upon the core aspects of our work and provide us with an opportunity to clarify our novelty and contributions in greater detail.
>
> **W1. On the novelty compared to WINA.**
> We respectfully clarify that our work presents a significant generalization over WINA, rather than an incremental improvement. WINA can be viewed as a constrained special case of our WiSparse framework where the layer-specific exponent $\alpha_l$ is manually fixed to 1.0 for all layers. Our key insight is that this fixed, one-size-fits-all approach is suboptimal because neural network layers are highly heterogeneous.
>
> **W2. On the experimental comparison with WINA.**
> Thank you for emphasizing the importance of WINA as a baseline. We completely agree and have included WINA in all our main experimental comparisons in Table 3.
>
> Our results consistently demonstrate that WiSparse significantly outperforms WINA, especially under high sparsity pressure.
> - On Llama-3.1-8B : At 50% sparsity, WiSparse is already 1.12 points ahead of WINA. At the more challenging 65% sparsity, this performance gap widens to 3.25 points, highlighting the robustness of our method.
> - On the larger Qwen2.5-32B model: The trend continues. At 65% sparsity, WiSparse achieves an average score of 62.91, which is 3.2 points higher than WINA.
>
> **W3. On insights about the learned $\alpha_l$ and the use of the L2 norm.**
> This is an excellent question that gets to the heart of our adaptive scoring mechanism. While we use the L2 norm as a base feature, the learned exponent $\alpha_l$ fundamentally transforms its role from a static measure into a dynamic, layer-adaptive component. The provided plots (See Fig.6 in Appendix) of learned $\alpha_l$ values for Llama-3.1-8B offer compelling insights, proving that a static norm is indeed inadequate.
> We observe distinct and systematic layer-type specific patterns:
> - Attention Projections: Within the attention block, the k_proj (key) layers consistently favor high $\alpha_l$ values (often > 1.0). This suggests that for these layers, the magnitude of weight columns is a very strong indicator of importance. In stark contrast, many v_proj (value) and o_proj (output) layers learn an $\alpha_l$ close to 0. This instructs the pruner to almost entirely ignore weight norms and rely on activation magnitudes, effectively making them behave like the TEAL method.
> - MLP Projections: In the MLP blocks, down_proj layers often settle near $\alpha_l$ = 1.0, aligning with WINA's heuristic. However, the up_proj and gate_proj layers exhibit significant variability, again demonstrating the need for layer-specific adaptation.

---

### Official Review · Reviewer_vQjd · 2025-11-14

**Soundness:** 3
**Presentation:** 2
**Contribution:** 2
**Rating:** 6
**Confidence:** 3

**Summary:**

This paper introduces a training-free activation sparsity framework called WiSparse. Unlike prior activation-only methods (e.g., CATS, TEAL), WiSparse combines activation magnitudes with precomputed weight norms via a layer-wise exponent and uses a coarse-to-fine sparsity allocation scheme (evolutionary search over block sparsities and greedy intra-block allocation). The method aims to better preserve accuracy at high sparsity (up to 50%).

**Strengths:**

- Clearly-motivated problem: shows empirical evidence that activation-only criteria can prune channels with small activations but very large weight columns, and that block-wise sparsity sensitivity is highly non-uniform.
- Mixed-granularity sparsity allocation (block-level evolutionary search + layer-level greedy search) is reasonable.
- Comprehensive empirical evaluation on three different 7–8B LLMs (Llama-3.1, Mistral, Qwen2.5) across multiple benchmarks, with consistent gains over strong training-free baselines (TEAL, R-Sparse), especially at 50% sparsity.
- Reports both FLOP reductions and real end-to-end throughput improvements on GPU, showing that sparsity translates into actual speedups.

**Weaknesses:**

- Conceptual novelty is somewhat limited relative to prior weight-aware sparsity (e.g., WINA) and activation-based methods (TEAL/R-Sparse).
- The calibration and search pipeline appears non-trivial, but the paper does not quantify its wall-clock overhead or resource requirements.
- Experiments are restricted to ~7–8B models and a single hardware setup; it is unclear how well WiSparse scales to larger models (e.g., 30B+) or different batch sizes.

**Questions:**

- How sensitive are the learned $α_ℓ$ and sparsity allocations to the choice and composition of the calibration set? Does performance degrade if evaluation tasks differ significantly from calibration tasks?
- Have you explored sparsity levels beyond 50% (e.g., 60–70%)? If so, how does WiSparse compare to TEAL/R-Sparse at those points, and where does accuracy begin to collapse?
- Can you quantify how much of the total inference time is spent computing scores/masks versus running sparse kernels, and how this scales with batch size and sequence length?
- Do you foresee any practical issues applying WiSparse to larger LLMs (e.g., 30B, 70B)? Any preliminary results or observations?

---

> ### Author Response · Authors · 2025-11-29
> **Response to reviewer vQjd**
>
> We sincerely thank you for your detailed and constructive feedback, which has helped us improve the quality of our paper. Below are our point-to-point responses to your comments.
>
> **W1. On conceptual novelty relative to prior work.**
> We respectfully disagree. While WINA introduced weight-awareness by combining activation magnitudes with weight column norms, our work advances this in two critical dimensions that WINA does not address:
> **Adaptive weight-activation interaction via learnable exponents ($\alpha_l$):** Unlike WINA's static use of L2 norms, we introduce a layer-specific exponent $\alpha_l$ (Eq. 4) that modulates the contribution of weight information per layer. This allows WiSparse to adapt to the heterogeneous characteristics of different layers. We optimize $\alpha_l$ via lightweight block-wise grid search (Algorithm 2), ensuring the weight-activation interaction is context-sensitive rather than fixed.
> **Mixed-granularity sparsity allocation:** Our coarse-to-fine allocation (Algorithms 3 & 4) assigns different sparsity budgets to different blocks and layers. This addresses a fundamental limitation: uniform sparsity is suboptimal (Sec. 4.1).
>
> **Overall, WINA is essentially a special case of our method when $\alpha_l=1$. Our framework is more general and flexible, enabling better adaptation to the heterogeneous characteristics across different model layers.**
>
> **W2. On the overhead of the calibration and search pipeline.**
> Thank you for pointing out this omission. We have now quantified the one-time calibration overhead:
> - Search cost breakdown (Llama-3.1-8B, single H20 GPU):
>     - Alpha search (Algorithm 2): ~30 minutes
>     - Block-level evolutionary search (Algorithm 3): ~2.5 hours
>     - Layer-level greedy refinement (Algorithm 4): ~1 hour
>     - Total calibration time: ~4 hours one-time cost
>
> **W3. On scaling to larger models (e.g., 30B+).**
> Thank you for this excellent suggestion. To demonstrate the scalability of WiSparse, we have conducted new experiments on a significantly larger model, Qwen2.5-32B, as shown in Table 3 of the revised manuscript. The results clearly show that WiSparse's superiority is not only maintained but often amplified on larger models.
>
> **Q1. On sensitivity to the calibration set.**
> | Calibration set | SIQA | GSM8K | WiC | HumanEval | MMLU | CSQA | Average |
> | :--- | :--- | :--- | :--- | :--- | :--- | :--- | :--- |
> | pileval | 43.81 | 77.06 | 52.35 | 60.37 | 65.58 | 77.56 | 62.79 |
> | pileval+code-alpaca | 42.89 | 76.12 | 50.47 | 64.63 | 64.87 | 77.24 | 62.70 |
> | pileval+code-alpaca+metamath | 43.14 | 78.77 | 50.63 | 65.85 | 65.23 | 77.81 | 63.57 |
>
> **Calibrated parameter $\alpha_l$ shows robustness:** In our experiments, $\alpha_l$ remains stable across different calibration sets.
> **Calibration set composition does impact final performance:** While $\alpha_l$ is stable, the choice of calibration tasks meaningfully affects downstream accuracy:
> - Adding code-alpaca improves humaneval performance (+4.26 points: 60.37→64.63)
> - Adding metamath enhances gsm8k results (+1.65 points: 77.06→78.77)
> - Using a diverse, all-round calibration set yields the best overall performance (avg: 63.57)
>
> **Q2. On performance at sparsity levels beyond 50%.**
> Yes, we have extended our evaluation to a 65% sparsity regime to analyze performance at these higher compression rates. The results are now included in Table 3. At these challenging sparsity levels, WiSparse demonstrates significantly more graceful degradation than all other methods:
> - On Llama-3.1-8B: At 65% sparsity, WiSparse achieves an average score of 46.08, outperforming the next-best baseline (WINA) by 3.25 points.
> - On Qwen2.5-32B: The performance gap widens on the larger model. WiSparse scores 62.91, a full 3.2 points ahead of the runner-up and maintaining strong capabilities, whereas other methods experience a more severe decline.
>
> Regarding the "collapse" point, for the 8B model, all methods see a sharp performance drop moving from 50% to 65% sparsity. However, for the 32B model, performance at 50% is still highly robust, and the more significant degradation begins past this point. In both scenarios, WiSparse consistently shows the highest resilience, confirming the effectiveness of its adaptive, mixed-granularity approach in extreme pruning conditions.

---

> > ### Author Response · Authors · 2025-11-29
> > **Continued response to reviewer vQjd**
> >
> > **Q3. On the inference overhead of computing scores/masks.**
> > As detailed in lines 241-246, both $g_i$ (weight column norms) and $\alpha_l$ (layer-specific exponents) are precomputed offline during calibration. At inference time, only the element-wise multiplication $|x_i| × (g_i)^\alpha_l$ is required. Our profiling shows this mask computation accounts for approximately 2% of total inference time, which is negligible compared to the substantial savings from skipping sparse matrix multiplications.
> >
> > **Q4. On practical issues and preliminary results for larger LLMs.**
> > This is an excellent forward-looking question. Our new experiments on the Qwen2.5-32B model (Table 2) serve as strong preliminary evidence for applying WiSparse to even larger models.
> >
> > Our key observation is that larger models exhibit greater resilience to sparsity. For example, when applying 65% sparsity:
> > - The Llama-3.1-8B model's average performance drops by ~19.5 points (from 65.57 to 46.08), a ~30% relative degradation.
> > - The Qwen2.5-32B model's performance drops by only ~8.9 points (from 71.83 to 62.91), a ~12% relative degradation.

---

### Author Response · Authors · 2025-11-29
**General Response**

We are deeply grateful to the reviewers for their insightful comments and valuable suggestions, which have been instrumental in helping us substantially strengthen our manuscript.

In direct response to the feedback, we have conducted extensive new experiments and added significant clarifications. The major updates include:

- **Scalability to Larger Models:** We have expanded our evaluation to a significantly larger 32B model (Qwen2.5-32B) to demonstrate the scalability and effectiveness of our approach, addressing a key suggestion from several reviewers.

- **Performance at Higher Sparsity:** We have investigated performance at a more challenging 65% sparsity level, revealing that the advantages of WiSparse become even more pronounced under extreme compression. Specifically, our method outperforms the second-best method by 3.19%-3.24% in average accuracy under the 65% setting.

- **Direct Comparison with WINA:** We have incorporated a direct and comprehensive comparison with WINA across all new experiments, confirming WiSparse's superior performance as requested. Specifically, our method achieves an average accuracy improvement of 3.19%-3.24% over WINA under the 65% setting.

The comprehensive results for these three crucial experimental updates are now consolidated in **Table 3 in the Appendix of the revised version**.

In addition, we have provided a new **layer-wise visualization and analysis of the $\alpha$ values** (Appendix, Fig. 6) to offer deeper insights into our adaptive scoring mechanism.

A revised manuscript has been uploaded, with all significant changes and new text marked in <span style="color: blue;">blue</span> for easy identification. We believe these additions, made in direct response to your comments, more robustly validate our contributions and clarify the novelty of our work. Detailed point-to-point responses are provided below.

Sincerely,

The Authors

---

### Meta-Review · Area_Chair_a5rT · 2025-12-19

**Summary:**

Reviewers discussed extensively on the degree of conceptual novelty beyond existing weight-aware sparsity methods, the cost–benefit trade-off introduced by the multi-stage calibration and search pipeline, and whether the reported accuracy gains justify the added complexity.

While reviewers generally agreed that the method is technically sound and empirically well executed, many viewed the core ideas as an incremental generalization of prior work rather than a fundamentally new sparsity principle.

The rebuttal strengthened the empirical scope by adding higher-sparsity regimes and larger models, but it did not fully resolve concerns about over-engineering, esp. the necessity of several design choices. Overall, the consensus was that the paper represents a solid piece of engineering work, but falls short of the conceptual novelty and clarity expected for ICLR acceptance.

**Reviewer Concerns:**

Several substantive concerns were partially addressed during rebuttal, including missing comparisons with WINA, evaluation at higher sparsity levels (e.g., 65%), and clarification of calibration overhead through explicit timing breakdowns. The authors also improved presentation clarity and expanded experiments to a 32B-scale model.

However, reviewers remain unconvinced regarding the limited conceptual separation from prior methods, the lack of a compelling argument for why the proposed evolutionary and mixed-granularity machinery is necessary over simpler alternatives, and whether the gains are commensurate with the added complexity. In addition, a key empirical observation (e.g., layer-wise sparsity sensitivity variation) is viewed as less a novel contribution, but more a re-demo of known phenomena.

**Reviewer Scores:**

With AC's most educated guess, reviewers who were initially negative would likely maintain their stance

---

### Decision · Program_Chairs · 2026-01-26

Reject